# Support Needs of Parents of Adolescents Abusing Substances in Selected Hospitals in Limpopo Province [note 1]

**DOI:** 10.3390/children10030552

**Published:** 2023-03-15

**Authors:** Lina Sebolaisi Hlahla, Charity Ngoatle, Tebogo Maria Mothiba

**Affiliations:** 1Department of Nursing Sciences, University of Limpopo, Polokwane, Sovenga 0727, South Africa; 2Faculty of Health Sciences, University of Limpopo, Polokwane, Sovenga 0727, South Africa

**Keywords:** support, parents, adolescent, substances abuse

## Abstract

Parents with adolescents who abuse substances need support. They have high stress levels and low quality of life compared to other parents. This is because they have unmet support needs, do not know what to do to help their adolescents, and are distressed. Most studies focus on the support needs of adolescents. Less is known about the specific support needs of their parents. This study explores the support needs of parents of adolescents abusing substances and being treated in five hospitals in Limpopo Province. A qualitative research approach was applied with an explorative, descriptive, and contextual design using semi-structured interviews to understand parents’ views. Data saturation was reached at the 14th parent. Data were analyzed using the Tesch method. The parents mainly wanted informational and emotional support. The study identified specific parent-related support needs and adolescent-related support needs. This study is the first to explore the support needs of parents of adolescents abusing substances in a South African rural context.

## 1. Introduction

Parents with adolescents abusing substances need support. They suffer stress, frustration and anger related to the abuse of substances by their adolescents. This is because parents are afraid that their adolescents may end up suffering mental illness related to substance abuse and later become a burden to the whole family [1]. Parents have a responsibility to manage their adolescents. They do that by shaping the behavior of their children, which is called parental control. That includes setting rules of conduct, supervising the adolescents, and disciplining them. Parents can also show warmth to their children by showing that they are involved in their lives emotionally and physically. Parental warmth and parental control have proven to have protective roles in reducing the abuse of substances by adolescents [2].

When adolescents abuse substances, parents are blamed by society. Parents also blame themselves. They are expected to cope with the consequences of their importance to their children. The results may range from depression and anxiety to humiliation and feelings of loss. They also suffer from guilt, worry, family conflicts, and shame. Parents further fail to cope with the stress and strain caused by the abuse of substances by adolescents [3].

There are negative impacts of adolescent substance abuse within a family. The use of substances by adolescents may lead to family disorganization, disharmony, family disruption, family conflicts, and financial burdens. Parents find adolescents who abuse substances challenging to manage. They go through different emotions, such as shame, guilt, anger, resentment, and depression. They are generally embarrassed or ashamed of their child’s substance abuse [4,5].

Parents are affected by the abuse of substances by their children. They suffer in silence with little or no support. Most substance abuse treatment centers treat people with substance abuse dependence, neglecting the families that brought them to the center. There is a need for support services to help parents cope with the challenges they endure as a result of their child’s substance abuse [6].

The inability to control adolescents abusing substances leads to parents feeling helpless. It leaves them despondent. They feel hopeless when they have made efforts such as talking to the adolescent about the dangers of taking substances, but their efforts could not yield better results. It is more frustrating for parents when they have taken their adolescents to the rehabilitations centers before but they continue abusing substances [7].

This study explores the support needs of parents of youths who are abusing substances in selected hospitals in Limpopo Province, South Africa. Currently, there are no support programs for the parents of youths who are abusing substances in Limpopo Province. Historically, the use of substances was associated with middle-class communities; however, because of availability of cheaper drugs such as cannabis, the abuse of substances is increasing in rural provinces such as Limpopo [8]. The report by the South African Community Epidemiology Network on Drug Use states that there has been an increase in the number of users seeking treatment for the abuse of substances by young people in the Northern Provinces of South Africa, such as Limpopo [9]. An increase in the abuse of substances by adolescents has put a burden on parents who have to deal with the consequences of the of the addiction of their adolescents on a daily basis [10]. Thus, the there is a need to support parents.

## 2. Methods

### 2.1. Research Design

This study adopted a qualitative research approach using an explorative, descriptive, and contextual design. The explorative design assisted in the collection of the views of parents regarding their support needs. An illustrative method was adopted to summarize and synthesize the support needs of parents of adolescents abusing substances in the context of the study, which was a selected hospital in Limpopo Province.

### 2.2. Study Setting

The study was conducted in the selected hospitals of Limpopo Province. Limpopo Province is located in the northern part of the Republic of South Africa. The reason for the study to be conducted in Limpopo Province is that there has been an increase in the abuse of substances by adolescents in Limpopo Province. The abuse of substances is different based on district. Based on the report from the Department of Social Development, 26% of adolescents in Limpopo Province between the ages of 11 and 20 abuse substances, with the commonly abused substances being cannabis at 49%, inhalants at 34%, and alcohol at 58%. The authors wanted to know the support needs of parents of children abusing substances. Limpopo Province has five districts: Mopani, Sekhukhune, Waterberg, Vhembe, and Capricorn. The regional hospitals used in this study are Letaba, Jane Furse, Mokopane hospital, Tshilidzini, and Mankweng. The wards chosen for this study were those for adolescents admitted because of problems related to substance abuse.

### 2.3. Population and Sampling

The population for this study was the parents of adolescents abusing substances in selected hospitals in Limpopo Province. The parents of adolescents using substances were accessed during visiting hours and when they brought their adolescents for consultations. Homogeneous purposive sampling strategies were applied to select the parents. This sampling method was used because it ensured that the parents were representative of the population and met the standards the authors desired. They had all the qualities the authors wanted: being the parent of the adolescent abusing substances and staying with the adolescent abusing substances.

### 2.4. Data Collection Procedures

Data were collected through semi-structured interviews which lasted for 30 to 60 min. The central question that guided the study was, “what are your support needs as the parent of an adolescent abusing substances”. As the parents answered the main question, the authors enquired further to obtain more information. In addition to taking field notes while gathering data, the authors used a voice recorder to capture their results. At the 14th parent, data saturation was attained. No incentives were offered to the parents for participation in the study. The 14 parents were either the father or the mother of the admitted adolescent. Table 1 presents the demographics of the parents.

## 3. Data Analysis

The following eight steps of Tesch’s data analysis were used to analyze data: Data were read by authors and converted to codes. The authors re-read the transcripts and analyzed them. There were checks made for duplication of codes and topics. Then, themes and subthemes were created. Data belonging to each theme and subtheme were assembled. The independent co-coder consulted to come to a consensus on themes and subthemes. The existing data were then recorded [11].

### 3.1. Measures to Ensure Trustworthiness

Measures to ensure trustworthiness were adhered to. Credibility was ensured through prolonged engagement with parents during data collection for two months until data saturation was reached. An audit trail was used to ensure confirmability. To properly understand the topic under research, the authors conducted recorded in-depth interviews and kept field notes. Verbatim transcripts were later written. Dependability was confirmed using an independent coder to reach a consensus regarding themes and subthemes. A full description of the research methodology used in this study assisted in achieving transferability.

### 3.2. Ethical Clearance and Ethical Considerations

Ethical approval for this study was received from the University of Limpopo’s Turfloop Research Ethics Committee (TREC), number: TREC/305/2018: PG. Officials were the Department of Health Limpopo Province (approval number LP 201902_009), the Department of Health district offices in Limpopo Province, and the CEOs from the hospitals selected. The parents were briefed about the study, and agreed to be part of the study voluntarily. The authors let the participants know they had the right to withdraw at any time without repercussions. The researchers were assured of confidentiality and anonymity. Parents signed a consent form before the interview began.

## 4. Results

The parents’ responses during data collection led to the development of the themes and subthemes tabulated below. Table 2 presents the themes and subthemes as they emerged from the study.


*Theme 1: Parent-related support needs*



*Subtheme 1.1. Need for knowledge related to substances and substance abuse*


The study found that the parents of adolescents abusing substances wanted to know about substance abuse. They were failing to cope with their adolescents using substances. They did not know what to do with their adolescents. The parents gave the following accounts describing their need for substance knowledge.


**Parent 4 said,**
*“I wish they could teach me about his substance abuse problem. Maybe they must also teach us their coping strategies with these many patients in the ward. I don’t know what to do with him when he is home. I need to learn more about substance abuse. Maybe if there are foods he must not eat, they must tell me about them. I want to learn as much as possible about his condition”.*

*(Parent 4, age 45, female)*


**
Parent 3 said,
**
“I need to know if my child is complicating after using substances and how I will help him. What kind of medications can I use for him to stop using substances? Again, how can I support him in his journey of quitting substances because we were not taught how to keep our kids who use substances? As a parent, I know nothing about this substance abuse. I get surprised when I see him high”.(Parent 3, age 39. female)

**
Parent 7 said,
**
“Maybe if the nurses and doctors can teach me how to handle him and help him stop using the substances, I will be a better person”.(Parent 7, 47. male)

**
Parent 9 said,
**
“My family does not know what to do, and I also do not know what to do”.(Parent 9, age 54, female)


*Subtheme 1.2: Need for feedback*


Parents reported that they needed feedback from healthcare providers. They needed some encouragement when they were doing well. When parents received feedback from healthcare providers, they became confident and less worried. This is supported by the statements below from parents:

**Parent 1 said,**
“They must greet us and tell us what is happening with the patient to make us feel better because I am always worried”.(Parent 1, age 32, female)

**
Parent 6 said,** “If they can just talk to us and reassure us that we are doing a good job taking care of the patient at home, that will be enough”.(Parent 6, age 40, female)

**
Parent 7 said,**
“the nurse needs to be kind to us as relatives. They must greet us and tell us what is happening with the patient to make us feel better because I am always worried”.(Parent 7, age 47, male)


*Subtheme 1.3: Need for support groups*


Parents reported that they needed support groups. This study revealed that parents knew about support groups and the impact they could have on their lives as a form of support. This was confirmed by the parents who gave the views below:

**
Parent 7 said,
**
“Maybe there can be a training that we attend every week. He must also be part of that training to learn *how to get out of substances*”.(Parent 7, age 47, male)

**
Parent 11
said,** “I heard in Gauteng there are some meetings for the parents of the patients who use substances. We don’t have much in this hospital. I wish we had such. That way, we will be quickly involved in the care of my son”.(Parent 11, age 44, female)

**
Parent 5
said,** “I am frustrated I do not know what we are dealing with currently. I can do some support programs for parents. I also wish we had aftercare where these patients are regularly met to see their progress, or the home-based carers must visit us often to check on us. Now have to stop working and come here like I am the one who is sick. I am not doing well”.(Parent 5, age 50, female)

**
Parent 13
said,** “Maybe they can also bring the parents who have been with children who abused substances to come and relate to how they manage their children who use substances. Maybe they can give us coping strategies”.(parent13, age 48, female)


*Theme 2: Adolescent-related support needs.*


Parents reported that the following factors could support them as they cared for adolescents abusing substances.


*Subtheme 2.1: Adolescents awareness programs related to substances.*


Parents wished that there could be awareness programs for adolescents abusing substances. Awareness programs serve as a way of sharing information about the dangers of substances to the adolescents. The claims below serve as evidence of what the parents said:

**
Parent 14,
**
“I wish they could bring a person who was once a substance abuser who can come and talk to my son about how he got out of substances”. (Parent 14, age 48, female)

**
Parent 6,
**
“I hope there could be programs for drug users like alcohol anonymous for alcoholics”.(Parent 6, age 37, female)

**
Parent 9
said,** “They must also teach my son to accept himself. He is always isolated”.(Parent 9, 54)

**
Parent 2,
**
“I wish they could talk to my son and tell him that substances are unsuitable and negatively affect his bright future.”(Parent 2, age 52, female)


*Subtheme: 2.2: Support groups for adolescents abusing substances*


Just as parents want support groups for themselves as the parents of adolescents abusing substances, they also suggested the need for support programs for adolescents abusing substances. The quotes below support their need for support programs:

**
Parent 9
said,** “Maybe if we can get programs regarding the adolescent who are using substances… the substances are busy destroying their bodies”.(Parent 9. Age 54, female)

**
Parent 6
said**, “I also wish we had aftercare where these patients are regularly being met to see their progress, or the home-based carers must visit us often to check on us. Now I had to stop working and come here like I was the one who was sick. I am not doing well”.(Parent 6, age 37, female)

**
Parent 7 said,** “There must also be community centers or programs that help during discharge. This thing takes the whole life to be truly liberated. At the hospital, we get help, but when we go home, nothing happens”.(parent 7, age 47, male)


*Subtheme 2.3: Availability of institutions for adolescents abusing substances*


Parents reported that the availability of institutions could serve as a support to them. This is because rehabilitation centers provide care and treatment for longer periods where adolescents will be separated from friends who might influence their substance abuse. This is supported by the following statements from study participants: 

**
Parent 14
said,** “I also hope they can refer us to the rehabilitation center. I do not know if the government can make such referrals to go there for some time. He must be away for some time because his friends are waiting for him when he returns home. He will soon go back to substance abuse. All I want is for him to be okay. I do not care how long he goes away”.(Parent 14, age 42, female) 

**
Parent 1 said,** “I wish the hospital could open a rehabilitation center or maybe write a letter for us to take him to the rehabilitation center. They must keep him in the hospital for a long time. They can also refer us to a psychologist because his substance abuse does not affect him alone. It affects us all as a family”.(Parent 1, age 32, female) 

**
Parent 12 said,** “There must also be community centers or programs that help during discharge. This takes the whole life to be truly liberated. At the hospital, we get help, but when we go home, nothing happens”.(Parent 12, age 60, female)

## 5. Discussion

This study aimed to explore the support needs of parents of adolescents abusing substances. There are two main themes that emerged from the study. The themes were parent-related support needs, which had three subthemes, and adolescent-related support, which had three subthemes.

The results indicate that parents need to know more about substances. Parental knowledge regarding substance abuse plays a role in reducing the abuse of substances by adolescents. When parents know about the abuse of substances, it is easy for them to discipline their children and communicate with them about the dangers of substance abuse. With high parental knowledge, there are low rates of substance abuse [2]. Obtaining information from healthcare providers is the right of parents and of adolescents abusing substances. It is, therefore, the duty of healthcare providers to share information with parents at every opportunity they have to be with the parents. The need for information by parents includes being given opportunities to ask questions and being allowed to be part of the decision-making process around the care of their adolescents [12,13,14]. The anxieties that parents have can be reduced when parents know about substance abuse. Healthcare providers should therefore use every opportunity they have to share information with parents.

The results of this study indicate that parents of adolescents abusing substances need feedback. They want to know whether they are doing well or not taking care of their adolescents using substances. This is supported by Davis-Strauss et al. [15], who stated that when healthcare professionals provide feedback to the parents, they make them feel safe, increasing their belief that parents want more coordinated and individualized care as well as to be reassured by the healthcare professionals in their parenting and caregiving during the transition of assuming the responsibility of taking care of the adolescents abusing substances. Parents prefer support from healthcare providers even though they have family support. Healthcare providers must make a conscious effort to show concern for parents and patients. Healthcare providers must conduct support visits to parents to encourage them to keep doing well. According to Loft et al. [16], a good relationship between parents and healthcare professionals improves the outcomes of care by the parents. Healthcare professionals need to show empathy to parents. They need to be able to talk to parents, encourage them to do well, and guide them where they fail so that they avoid the stigma related to being a parent of a substance abuser.

The parents, as participants in this study, showed that they need support groups. Support group interventions can assist parents in obtaining social support from fellow parents who undergo similar situations. Support groups provide peer support with the help of a facilitator who can be a healthcare professional. It can also occur when parents are matched with other parents going through the same problem as caring for the youth who are abusing substances [17]. This is because, according to Caldarera et al. [18], support groups are a valuable tool for working closely with parents. The participation of parents in support groups gives parents a better understanding of their situations. They can understand their emotional challenges and their children’s challenges. Support groups have also been found to improve parent–child relationships. With support groups, parents can gain confidence and feel less lonely. It also enhances the well-being of parents by giving them optimistic hope. The primary purpose of support groups is to encourage parents to share their consent, knowledge, and achievements in a mutually supportive network. The study by Jackson et al. [19] reported that support groups can be helpful for the parents of youth who are abusing substances. However, the support groups’ design and content are essential to the parents. It is necessary for the healthcare workers to conduct the support group in such a way that they are helpful to parents. Support groups can be done virtually and in person. An online support group is an alternative for parents, as many parents go online to seek information and support [20].

The study results have indicated that parents want to be supported through awareness programs for youth who are abusing substances. Adolescents are reluctant to seek help for several reasons, such as a lack of knowledge regarding substance abuse, poor understanding of the dangers of substances, and where to find help if addicted. They believe they can handle their problems. The other factor can be previous bad experiences [15]. There is a need for substance abuse awareness campaigns in communities to enlighten adolescents about the adverse effects of substance abuse. Awareness programs will capacitate adolescents with all the information they need regarding substances [21]. Awareness programs must include factors to increase youth involvement and participation. When awareness programs encourage youth involvement, they build relevance for youth and can eliminate many obstacles that may make them uninterested in such programs. Part of n awareness program’s content should be to make communities aware that their community beliefs and norms can contribute to the abuse of substances. Communities should be mindful of the effects of standards and beliefs, especially if they lead to substance abuse. Substance abuse awareness campaigns in congregations will enlighten the community about the adverse impact of substance abuse [15,21]. The awareness program can assist in the spread of knowledge regarding substance abuse to adolescents and slow down the abuse of substances by the adolescents.

The study findings revealed that parents want a space where adolescents can meet and share their experiences with substance abuse. There is a need for support groups for adolescents. These adolescent support groups can serve as a support to the parents. According to Sellers et al. [22], peer support has the potential to enhance recovery. The outcomes can improve outcomes for both the youth and their families. Support groups are associated with high self-esteem among youth, low anxiety and depression, and better sleep. Support is positively related to improved mental and physical health [23]. Peer support groups may be able to provide support to youth who are abusing substances and can fulfill the function of informational and emotional support. Peer support can alleviate the youth’s mental health stress and provide social support. It can also motivate the youth to take better care of themselves. There will be better coping mechanisms among the youth [24]. Support groups can assist the youth to better take care of themselves and decrease the likelihood of relapse. As support groups help with coping mechanisms, they can also assist with buffering against the stigma related to substance abuse and its consequences.

The study revealed that parents want institutions where they can send their adolescents for further treatment and rehabilitation. An alternative approach to dealing with substance abuse is required. Establishing rehabilitation centers that can be accessed by all members of society from different socioeconomic statuses is necessary. Accessibility to such a center can help reduce the abuse of substances by the youth and serve as a support system for the youth. The advantage of rehabilitation centers is that they provide multidisciplinary assessments for creating an individualized plan [25,26]. According to Mohasoa and Mokoena [27], there is a need for services that can help with aftercare for substance abuse. Aftercare services are associated with reduced relapses, and they equip substance abusers with skills to maintain treatment gains and sobriety.

Based on the discussions provided, it is recommended that parents be supported, and the support should be offered from healthcare institutions—healthcare professionals need to ensure that the parents of admitted adolescents abusing substances are given attention. Healthcare professionals should create a time to talk about how they are doing as parents of youth using substances. Healthcare providers must use every opportunity they have to give the parents information. Information can either be on substance abuse or the progress of their adolescent in the hospital. If healthcare providers have no time to communicate with the parents due to workload, they can make sure that within hospitals there are posters and information boards addressing substance abuse issues. Within the wards, where parents visit their adolescents, there must be information pamphlets addressing substance abuse.

It is also recommended that health institutions develop programs where the parents are educated on substances and substance abuse in a formal way. The agenda should cover the parents’ support needs, including sharing information regarding substance abuse. These kinds of programs can start from community clinics through to health education and awareness campaigns to prevent hospital admissions. The healthcare professional can support parents by organizing support groups in which parents will have a safe space to share their experiences in dealing with adolescents abusing substances.

## 6. Limitations

This study was conducted at selected hospitals in Limpopo Province, so the study’s findings cannot be generalized to other settings.

## 7. Conclusions

The study investigated the support needs of parents of adolescents abusing substances. Parents indicated that they could feel supported when given information regarding the implications; they also wanted to know if what they were doing in taking care of their adolescents abusing substances was correct; and they also need support groups. They also gave an account of how they want to be supported through the care they wanted their adolescents to receive. Healthcare professionals need to focus on parents as much as they care for adolescents abusing substances. There should be information sessions between parents and the healthcare providers. The development of support groups for parents and adolescents using substances can assist in reducing the pressure the parent goes through when taking care of the adolescents abusing substances. Implementing the suggested support needs of parents will assist in managing the stress and anxiety of the parents who are managing their adolescents abusing substances. 

## Figures and Tables

**Table 1 children-10-00552-t001:** Demographics.

Parents	Number of Participants	n = 14
Gender	Female	12
	Males	2
Residential	Rural	12
	Semi-rural	2
	urban	0
Level of education	Primary school	2
	Secondary school	10
	Tertiary	2
Employment	Employed	2
	Unemployed	12

**Table 2 children-10-00552-t002:** Themes and subthemes.

Themes	Subthemes
Parent-related support needs	1.1 Need for knowledge related to substances and substance abuse
	1.2 Need for feedback
	1.3 Need for support groups
2. Adolescent-related support needs	2.1 Adolescent awareness programs related to substances
	2.2 Support groups for adolescents abusing substances
	2.3 Availability of institutions for adolescents abusing substances

## Data Availability

Not applicable.

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
