# Peer review of "Support Needs of Parents of Adolescents Abusing Substances in Selected Hospitals in Limpopo Province†"

_children, 2023, doi:10.3390/children10030552_

Round 1

Reviewer 1 Report

I have few questions to address, in particular about the local setting and the drugs used.

It is not clear to me if the 14 parents were mothers and fathers of 7 adolescents, or were they single parents?

What were the drugs used by the selected adolescents. What is the prevalence of drugs in the local setting and the Country?

Were the parents aware of the kind of substances used by their sons/daughters? Different strategies are often needed to cope with different classes of abuse, either stimulants or sedatives.

Why saturation was decided at 14 parents? 

How long did it take to interview those 14 parents?

Are the local police agencies informing citizens on what is going on the illicit market?

The language and the style need a mild revision. e.g. The other parent  said =  parent 3 said.

Can you provide a table with the characteristics of subjects/parents? ì include in the study? I could not find it.

Author Response

The comments effected, see the attached document.

Reviewer 2 Report

The purpose of the study was to qualitatively assess the support needs of parents with children receiving treatment for substance use disorder.  Findings indicated that parents expressed the need for more education and affirmation from health care providers and to implement support groups.  While the study does seem to make a novel contribution to the literature, I have concerns as it is lacking in several areas which include providing more information on the context of the study in the introduction and methods sections.  As a qualitative researcher, I also have some concerns about the structuring/presentation of the results section. 

Introduction:

A better contextualization of the sources being cited is needed.  For example, the first source cited (Collins et al., 2020) is used to support the argument that parents of adolescents with substance use disorders (SUDs) have unmet support needs.  However, this source is about parents whose children have life-limiting conditions (i.e., terminal) and not a SUD.  This occurs again with source seven (Aoun et al., 2020) being used to support the assertion that parents “can feel empowered if they get support and their children improve from substance use disorder”. This source is about parents whose children have cancer.  The authors should carefully consider how they are using their sources.

The authors could consider better situating the study in its geographic context, especially for the international reader.  Statistics about substance use rates among adolescents in the region/country could be provided.  Also a description of the types of services available could be useful.  How accessible are treatment programs? Do hospitals have any support systems in place at all?

The authors can provide more of an attempt to characterize the novel contributions of the study based on the geographic context and/or the context of parents of adolescents with SUDs.  This would require more of a literature review of the types of studies that have been conducted and among which populations. For example specifying that many studies on the need for support are conducted among parents of children with cancer or terminal conditions, etc…..

Methods:

The authors could provide more information about the population and sampling procedure and data collection procedures.  What was the response rate? Were there any incentives provided? Were the interviews conducted with one parent or both parents or a combination? What was the interview length time range?

Measures to ensure trustworthiness section- Can the authors clarify what some of this information means? What does prolonged engagement with parents mean? What is an audit trail? Why was the independent coder process not also mentioned in the data analysis section? An independent coder that was not an author?

Results:

Can the authors provide a justification of why sociodemographic characteristics were not obtained or described (i.e., sample characteristics table)?  It seems there would be differences in support needs by factors such as education or income level or even gender (mothers vs fathers). Not providing this information is consistent with a lack of context overall throughout the manuscript. 

The results section is lacking in a comprehensive analysis or presentation style that is consistent with qualitative studies.  I suggest the authors consult with a qualitative researcher and review qualitative literature to enhance this section. The qualitative studies cited below do not directly relate to the topic of parents needing support for children who abuse substances but they do relate to various forms of parental support:

Foster, K., Mitchell, R., Young, A., Van, C., & Curtis, K. (2019). Parent experiences and psychosocial support needs 6 months following paediatric critical injury: A qualitative study. Injury, 50(5), 1082–1088. https://doi.org/10.1016/j.injury.2019.01.004

Brown J. (2018). Parents' experiences of their adolescent's mental health treatment: Helplessness or agency-based hope. Clinical child psychology and psychiatry, 23(4), 644–662. https://doi.org/10.1177/1359104518778330

Bektas, G., Boelsma, F., Wesdorp, C. L., Seidell, J. C., Baur, V. E., & Dijkstra, S. C. (2021). Supporting parents and healthy behaviours through parent-child meetings - a qualitative study in the Netherlands. BMC public health, 21(1), 1169. https://doi.org/10.1186/s12889-021-11248-z

Sullivan, L., Wysong, M., & Yang, J. (2022). Concussion Recovery in Children and Adolescents: A Qualitative Study of Parents' Experiences. The Journal of school health, 92(2), 132–139. https://doi.org/10.1111/josh.13114

Discussion:

Some parts of the discussion appear to be repetitive.  There could be more of an attempt to synthesize information rather than dedicating one paragraph per citation. 

Author Response

The comments affected see the attached document as well as the corrected manuscript

Reviewer 3 Report

The manuscript, although interesting, does not differ much from other studies present in literature both in terms of structure and in terms of the results obtained. From all the study results presented in other manuscript , it is clear that parents with adolescent abusing substances need professional assistance and support as evidenced by the challenges faced in terms of promoting, maintaining and restoring their mental healt. The authors should therefore better explain the differences that their study presents compared to the other studies reported in the literature since the results also in this case show the need to provide informational and emotional support to the parents of adolescents who use substances, also indicating the type of drugs that adolescents use

Author Response

Thank you for your comments; see the attached document and the manuscript.

Round 2

Reviewer 2 Report

See original comment:

The authors can provide more of an attempt to characterize the novel contributions of the study based on the geographic context and/or the context of parents of adolescents with SUDs.  This would require more of a literature review of the types of studies that have been conducted and among which populations. For example specifying that many studies on the need for support are conducted among parents of children with cancer or terminal conditions, etc…..

This section is still lacking in substance and justification of the novel contributions of the manuscript.

See original comment:

The results section is lacking in a comprehensive analysis or presentation style that is consistent with qualitative studies.  I suggest the authors consult with a qualitative researcher and review qualitative literature to enhance this section.

This section is still not written in a way that qualitative research should be written.

Author Response

I tried to address the reviewer's comments, hoping for a positive response. 

Reviewer 3 Report

 The  manuscripthas been sufficiently improved to warrant publication in Children

Author Response

slight improvements made to the manuscript.